# Microbial Evolution in Artisanal Pecorino-like Cheeses Produced from Two Farms Managing Two Different Breeds of Sheep (Comisana and Lacaune)

**DOI:** 10.3390/foods13111728

**Published:** 2024-05-31

**Authors:** Francesca Luziatelli, Renée Abou Jaoudé, Francesca Melini, Valentina Melini, Maurizio Ruzzi

**Affiliations:** 1Department for Innovation in Biological, Agro-Food and Forest Systems (DIBAF), University of Tuscia, 01100 Viterbo, Italy; raj@unitus.it (R.A.J.); ruzzi@unitus.it (M.R.); 2Council for Agricultural Research and Economics (CREA), Research Centre for Food and Nutrition, 00178 Rome, Italy; francesca.melini@crea.gov.it (F.M.); valentina.melini@crea.gov.it (V.M.)

**Keywords:** Comisana breed, Lacaune breed, sheep milk, microbiome, Next-Generation Sequencing (NGS), Pecorino cheese

## Abstract

“Pecorino” is a typical semi-hard cheese obtained with raw or heat-treated sheep milk using procedures to valorize the raw material’s chemical and microbiological properties. In the present study, using a high-throughput method of 16S rRNA gene sequencing, we assessed the evolution of the microbiome composition from milk to Pecorino-like cheese in artisanal processes using milk from Comisana and Lacaune sheep breeds. The comparative analysis of the bacterial community composition revealed significant differences in the presence and abundance of specific taxa in the milk microbiomes of the Comisana and Lacaune breeds. Next-Generation Sequencing (NGS) analysis also revealed differences in the curd microbiomes related to dairy farming practices, which have a relevant effect on the final structure of the Pecorino cheese microbiome.

## 1. Introduction

Fermented foods have been a significant part of the human diet since prehistoric times [1]. These foods benefit consumers through nutritional content, high digestibility and microbial stability and represent the means of storage of humanity’s oldest foods [2]. Fermented foods are characterized by microorganisms, which define the product’s organoleptic characteristics and provide beneficial components such as probiotics and antioxidant and anti-pathogenic compounds [3]. They may also contain prebiotics that promote beneficial bacteria growth and, therefore, can modulate the host microbiota [4].

Among fermented foods, cheese represents a key component of the human diet, and its consumption is increasing worldwide [5,6]. Pecorino cheese is commonly referred to as a variety of hard and semi-hard cheese obtained with sheep’s milk by traditional procedures [7]. Refrigerated milk is filtered, heat treated (62–68 °C for 15 s) and inoculated with native milk ferment cultures. The milk curdles between 38 °C and 40 °C with lamb’s rennet crust. The cooking of the curd must be performed at a temperature between 45 and 48 °C [8].

Italy is well known for producing many “Pecorino” and other sheep milk cheeses [7,9,10]. Among them, Pecorino Romano is one of the most important Italian PDO cheeses, of which more than 32.6 tons was produced in 2022 [11]. Besides these PDO cheeses, many Italian Pecorino-like non-PDO cheeses are manufactured in small artisanal farms following traditional methods. These artisanal products are appreciated for their distinctive traits linked to the production environment and the milk’s microbial biodiversity.

Raw milk artisanal cheeses convey ideas of tradition and culture, mainly for countries such as France and Italy [12], to such an extent that cheese tourism is seen as a possible development perspective in rural, mountain and natural remote areas [13]. Moreover, raw milk cheeses have been associated with a complex profile of volatile compounds and highly sensorial attributes, conferring unique organoleptic properties [14] compared to processed cheeses, which show a less intense flavor and ripen more quickly [15,16].

Although the organoleptic quality of artisanal cheeses produced using raw milk and natural curd is superior to that of the most widespread pasteurized milk cheese, these products may pose a threat to consumers’ health, and therefore, their safety should be carefully assessed to protect the producer and consumer interests [17,18]. Thus, monitoring microbiota composition and its evolution during fermentation and ripening is crucial for obtaining products with optimal sensory properties and safety characteristics [19]. Much effort has been put into investigating raw milk microbial communities to improve cheese production and safety due to their importance to the world’s population [20,21].

In recent years, high-throughput sequencing (HTS) of 16 rDNA gene amplicons has been widely used to investigate the evolution of the microbiome during the fermentation process [22]. This method overcomes the limitations of culture methods and permits the study of the microbial community profile and taxonomic evolution across space and time in dairy products [15,23,24,25,26]. Several factors, including animal breed and farming practice, can affect the structure of the milk microbiome [27,28].

The main aims of this study were to assess the microbiota diversity in raw milk, curd and Pecorino-like cheese of two different sheep breeds, Comisana (CSB) and Lacaune (LSB), and to compare the microbiome evolution during the cheesemaking process using high-throughput sequencing of the 16S rRNA gene.

## 2. Materials and Methods

### 2.1. Sample Collection

Milk, curd and cheese samples were collected from two dairy farms in the Amaseno Valley in the Province of Frosinone (Central Italy): one raised the Comisana sheep breed, (CSB) and the other raised the Lacaune sheep breed (LSB). Each farm used raw milk and residual whey native starter cultures from Ricotta (whey) cheese manufacturing (*scotta-innesto*) in the Pecorino-like cheesemaking process. Two independent bulk milk samples of 500 mL and a representative amount of curd (50 g) and middle-aged (20 days) Pecorino-like cheese (1/5 of a 250 g cheese form) were sampled for each farm. All samples were transported immediately, at 4 °C, to the laboratory. Curd and cheese samples were divided into several aliquots (about 0.5 g each). An amount of 50 mL of milk was added with 50 μL of 0.5 M ethylenediaminetetraacetic acid (EDTA) with a pH of 8.0 and pretreated to reduce fat content by centrifugation for 10 min at 350× *g* and 4 °C, as Luziatelli et al. described [29]. Total cells (somatic and bacterial) were recovered by centrifugation for 10 min at 11,200× *g* and 4 °C. All materials were stored at −20 °C until DNA extraction.

### 2.2. DNA Extraction and Purification

Total DNA was extracted from raw milk, curd and cheese samples using a commercially available kit (DNeasy Blood & Tissue kit, Qiagen GmbH, Hilden, Germany) with the following modifications. The cellular pellet, obtained from pretreated milk samples stored at −20 °C, was resuspended in 5 mL of saline phosphate buffer (PBS) containing 10 μL of 0.5 M EDTA with a pH of 8.0 and centrifuged at 11,200× *g* for 10 min at 4 °C. The resulting pellet was resuspended in 5 mL of saline buffer containing 1% Triton X-100 and incubated for 2 h at 37 °C to lysate the somatic cells. The suspension was centrifuged at 11,200× *g* for 10 min at 4 °C. The bacterial pellet obtained was resuspended in 180 μL of enzymatic lysis buffer and treated as described in the DNeasy Blood & Tissue kit instructions.

To purify total DNA from curd and cheese, samples (100 and 250 mg, respectively) were homogenized and treated as described for milk samples.

### 2.3. DNA Quantification

DNA was quantified using the Qubit^®^ fluorometer 3.0 with the Qubit™ dsDNA HS Assay Kit (Thermo Fisher Scientific, Rodano [MI], Italy).

### 2.4. Next-Generation Sequencing and Data Analysis

The metagenome was amplified for the V3-V4 region of the 16S rRNA gene using the primer pairs 343F (5′-TAC GGR AGG CAG CAG-3′) and 802R (5′-TACNVGGGTWT CTA ATC C-3′).

Sequencing was performed on the Illumina MiSeq version 3 sequencing platform system in 300-nucleotide (nt) paired-end mode.

Run statistics were determined using CLC Genomics Workbench 12 (Qiagen GmbH). The Illumina-generated reads were demultiplexed, quality filtered and analyzed using the “Quantitative Insights Into Microbial Ecology” (QIIME) pipeline [30]. Operational Taxonomic Units (OTUs) were assigned to the reads using an open reference approach with the UCLUST algorithm [31] against the SILVA database release 138.1 (https://www.arb-silva.de/ (accessed on 1 December 2023)) clustered at 97% identity [32].

For the microbiome definition, the OTU data obtained from each sample’s replicate were combined and used to describe the most abundant bacteria in raw milk, curd and cheese samples. The resulting number of OTUs was converted into percentage abundance in Microsoft Excel (Microsoft Corp., Redmond, WA, USA) and used for comparative analysis.

Data were processed and visualized using Past (Paleontological Statistics) statistical software version 4.10 [33]. Sample datasets were compared using principal component analysis (PCA) with Bray–Curtis similarity and hierarchical clustering, as described by Luziatelli et al. [34].

## 3. Results

The Next-Generation Sequencing analysis of the 16S V3-V4 region generated reads between 28,584 (curd_1; CSB) and 56,259 (cheese_1; CSB; Appendix A). Approximately 58–85% of raw reads per sample passed the merging, trimming and chimera filtering steps and were analyzed with QIIME2 [35] (Appendix A).

The comparative analysis of the different microbiomes indicated that the number of OTUs per sample differed in the two sheep breeds. As shown in Appendix A, in the CBS microbiomes, the number of OTUs increased from 50 (milk) to 207 (curd), whereas in the LSB microbiomes, the OTU number varied between 66 (milk) and 165 (curd). Differences in the OTU abundance were also observed in the corresponding cheese samples: 19 (CSB)—54 (LSB; Appendix A). In Appendix A, the taxonomic assignment of each OTU is reported using the BLAST search against the SILVA database.

The rarefaction plots of the 16S rRNA datasets showed that all curves reached a plateau (Figure 1), indicating that the datasets represent the microbial community well.

### 3.1. Sheep Milk Microbiota

The comparative analysis of the milk microbiomes showed that the CSB and LSB bacterial communities clustered separately (Figure 2, Panel A), with principal coordinate 1 accounting for 100% of the total variance. The Venn diagram constructed using the OTU taxonomy data (Figure 2, Panel B) revealed the presence of 25 OTUs shared in both milk samples. These shared taxa represented 50% of the total CSB milk microbiome and 38% of the entire LSB milk microbial community (Figure 2, Panel B).

As shown in Appendix A, the OTUs shared between the two milk datasets were affiliated with 6 phyla and 19 families. About two-thirds of these 25 OTUs (16) belonged to *Proteobacteria*, with *Pseudomonadaceae* (4 OTUs, IDs: 332, 333, 338 and 340) and *Xanthomonadaceae* (3 OTUs, IDs: 347, 348 and 350) as the more abundant families. About 25% of the shared OTUs were equally distributed between *Actinobacteria* (3 OTUs) and *Firmicutes* (3 OTUs), with *Streptococcaceae* (2 OTUs, IDs: 156 and 159) as the more abundant family in the latter taxa.

We used principal component analysis (PCA) of the two datasets at the family (Figure 3, Panel A) and OTU (Figure 3, Panel B) levels to evaluate the main differences in the milk microbiomes of the two sheep breeds. This analysis revealed the presence of four families (*Pseudomonadaceae*, *Xanthomonadaceae*, *Enterobacteriaceae* and *Streptococcaceae*), whose abundance significantly varied among the two milk samples (Figure 3, Panel A). In the CSB milk samples, about 90% of the total OTUs were equally distributed between *Pseudomonadaceae* and *Xanthomonadaceae*. In contrast, the OTUs affiliated with these two families in the LSB milk samples were about 84% of the total reads. *Pseudomonadaceae*-affiliated OTUs represented approximately 60% of the total OTUs (Appendix A). PCA analysis also revealed that *Enterobacteriaceae* and *Streptococcaceae* were differentially abundant in the milk samples of the two sheep breeds (Figure 3, Panel A). *Enterobacteriaceae* was about 8-fold higher in LSB vs. CSB, whereas *Streptococcaceae* was 7.7-fold more abundant in CSB than that in LSB milk samples (Appendix A).

The analysis at the OTU level (Figure 3, Panel B) revealed that the differences at a family level were associated with the relative abundance of five different OTUs. OTUs 348 and 350, affiliated with *Xanthomonadaceae* (*Stenotrophomonas* spp.), and OTU 159, belonging to *Streptococcaceae* (*Streptococcus* sp.), were more abundant in the CSB microbiome. OTU IDs 340 and 307, belonging to *Pseudomonadaceae* (*Pseudomonas veronii*) and *Enterobacteriaceae* (unknown species), respectively, were more abundant in the LSB milk microbiota (Figure 3, Panel B).

### 3.2. Curd Microbiota

PCA constructed with all CSB and LSB datasets (raw milk, curd and Pecorino-like cheese) revealed that the differences between the two curd microbiotas were broader than those observed in a pairwise comparison between milk and cheese samples (Figure 4).

As reported in Appendix A, the number of OTUs in the raw curd samples was about 4.1 and 2.5 times higher than those in the corresponding CSB and LSB milk samples.

The Venn diagram constructed using the OTU taxonomy data revealed the presence of 67 OTUs shared in both curd samples, representing 32.4 and 34.3% of the total curd microbiome in the CSB and LSB samples, respectively (Figure 5, Panel A). The shared OTUs, affiliated with 37 different families, represented 93.4% (CSB) and 43.6% (LSB) of the total curd OTUs (Appendix A).

PCA at the OTU level indicated that the main differences between the curd microbiome datasets were associated with nine OTUs, two of which (OTU 79 and 271) were absent in the curd-shared taxa. OTU ID numbers 315 (*Serratia* sp.) and 337 (*Pseudomonas fragi*) were the most abundant in the CSB curd samples (Figure 5), representing about 4.3 and 80.5% of the total OTUs, respectively (Appendix A). OTU ID numbers 79 (*Flavobacterium frigidarium*), 156 (*Lactococcus* sp.), 271 (*Comamonas* sp.), 324 (*Acinetobacter* sp.), 326 (*Acinetobacter johsonii*), 328 (*Enhydrobacter* sp.) and 340 (*P. veronii*) were highly represented in LSB curd samples, with *F. frigidarium* (OTU 79) as the most abundant OTU (37.6%; Figure 5 and Appendix A).

### 3.3. Cheese Microbiota

The data reported in Appendix A indicate a 2.8-fold difference in the OTUs of the two cheese microbiomes (19 CSB vs. 54 LSB). In the two datasets, the number of OTUs whose abundance was higher than 1% was five (CSB) and eight (LSB). The relative abundance of these OTUs was 99 (CSB) and 92% (LSB).

PCA indicated that the differences between the CSB and LSB milk datasets were comparable with those at the cheese level (Figure 4).

#### 3.3.1. Influence of the Milk Microbiome

We used the OTU taxonomy datasets to construct the Venn diagrams reported in Figure 6 to describe the microbiome evolution during cheesemaking. This analysis revealed the presence of 9 (CSB) and 14 (LSB) shared OTUs between the milk and cheese datasets. About 80% of these OTUs were affiliated with *Proteobacteria* and *Firmicutes* and included, together with minor taxa, the *Pseudomonas veronii*-affiliated OTU 340 (the most abundant OTU in the milk dataset) and *Lactococcus* sp.-affiliated OTU 156 (the most abundant taxon in cheese datasets; Appendix A).

Analyzing the relative abundance of the shared OTUs between the milk and cheese microbiomes in the CSB datasets, we identified four OTUs whose abundance increased in cheese, four OTUs whose abundance was higher in milk and one OTU whose relative abundance remained constant during the cheesemaking process (Appendix A).

In the LSB datasets, the abundance of four of the shared OTUs increased from milk to cheese, five OTUs were more abundant in the milk microbiome, and three OTUs remained constant in their relative abundance in the two microbiomes (Appendix A).

In detail, the relative abundance of *P. veronii* OTU 340 significantly decreased from 43.8% (milk) to 0.02% (cheese) in CSB and from 58.9 (milk) to 0.04 (cheese) in the LSB datasets. In contrast, the relative abundance of *Lactococcus* sp. OTU 156 increased 66-fold in the LSB cheese (from 1.02 to 67.2% of the total OTUs) and 542-fold in the CSB samples (from 0.1 to 54.2% of the total OTUs; Appendix A).

The analysis of the LAB population showed that four OTUs related to four different genera (*Lactobacillus*, *Lactococcus*, *Streptococcus* and *Leuconostoc*) were shared between milk and the corresponding cheese (Appendix A). Notably, three OTUs were part of the shared milk taxa (*Lactobacillus zeae* OTU 153, *Lactococcus* sp. OTU 156 and *Streptococcus* sp. OTU 159; Appendix A). The OTU affiliated with the *Leuconostoc* genus differed in the two datasets: ID 155 in CSB and ID 154 in LSB.

Except for OTU 159 in the CSB milk samples, the other OTUs belonging to the LAB category increased during cheesemaking (Appendix A).

The Venn diagrams also show the presence of 7 and 11 unique OTUs in Pecorino-like cheese from CSB and LSB, respectively (Figure 6).

As shown in Appendix A, Panel A, in CSB Pecorino-like cheese, most of them (four out of seven) were affiliated with *Lactobacillaceae* (two OTUs), *Streptococcaceae* (one OTU) and an unclassified family of the *Lactobacillales* order (one OTU). The remaining OTUs were affiliated with three families: *Bifidobacteriaceae*, *Clostridiaceae* (*Clostridium perfringens*) and *Enterobacteriaceae* (*Rahnella aquatilis*).

In the LSB cheese sample, the 11 unique OTUs belonged to 10 families, among which the most represented was *Enterobacteriaceae* (*Citrobacter* sp.; 2 OTUs; Appendix A, Panel B).

The remaining OTUs were associated with nine different taxa: (1) *Micrococcaceae* (*Rhotia* sp.); (2) *Pasteurellaceae* (*Mannheimia* sp.); (3) *Pseudoaltermonadaceae* (*Pseudoalteromonas* sp.); (4) *Lactobacillaceae* (*Lactobacillus paralimentarus*); and (5) *Streptococcaceae* (*Lactococcus gavieae*), as well as unclassified species belonging to (6) *Caronobacteriaceae*; (7) *Dermabacteraceae*; (8) *Flavobacteriales*; and (9) *Lactobacillales* (Appendix A, Panel B).

#### 3.3.2. Influence of the Curd Microbiome

The OTUs shared between curd and cheese can be divided into two groups: those common to milk, curd and cheese (described above) and those shared only between curd and cheese. To analyze the potential role of the latter group of OTUs (3 in the CSB datasets and 29 in the LSB datasets; Figure 6), we arbitrarily clustered these OTUs into four categories, as described by Secchi et al. [34]: LAB and other probiotics (LAB), spoilage (SP), pathogenic (P) and other bacteria (O). In the CSB samples, one of the shared OTUs (ID 127) belonged to the pathogenic/spoilage category, and two (ID 315 and 337) belonged to the spoilage category (Appendix A).

The LSB curd and cheese samples also revealed OTU 127, affiliated with *Staphylococcus* (Appendix A). Its abundance decreased about 6-fold (from 0.06 to 0.01% of the total OTUs) from curd to cheese in the CSB samples and by only 20% in the corresponding LSB samples (from 1.54 to 1.32% of the total OTUs).

The abundance of the shared SP taxa decreased in the CSB samples from 84.81% (curd) to 39.55% (cheese) of the total OTUs. Interestingly, the trend of these two spoilage-related OTUs followed a different pattern: the abundance of OTU 337 (*P. fragi*) decreased about 20-fold (from 65 to 3% of the total OTUs), whereas the abundance of OTU 315 (*Serratia* sp.) increased approximately 10-fold (from 3 to 29% of the total OTUs).

The abundance of OTU 126, affiliated with *Macrococcus* sp. (putative pathogenic/spoilage taxon), increased about 4-fold (from 0.05 to 0.20% of the total OTUs) in LSB samples during the cheesemaking process. In the same samples, the SP category comprised 15 shared OTUs belonging to four phyla: *Proteobacteria*, which was the most representative phylum (10 OTUs); *Bacteroidetes* (2 OTUs), *Firmicutes* (2 OTUs) and *Actinobacteria* (1 OTU; Appendix A). Most of the taxa included psychrotrophic environmental bacteria, generally present in the soil, water and air [35]. The total abundance of these SP OTUs decreased about 25-fold (from 63.9 to 2.6% of the total OTUs) during the LSB cheesemaking process. In particular, OTU 79, affiliated with *F. frigidarium*, represented about two-thirds of the total SP-related OTUs and decreased about 125-fold (from 37.6 to 0.3% of the total OTUs) from the curd samples to the cheese samples (Appendix A).

In the LSB samples, OTUs belonging to the LAB and O categories were also revealed (Appendix A). LAB comprised nine OTUs belonging to five families and nine species. Half of these OTUs were affiliated with *Lactobacillus* (two OTUs) and *Streptococcus* (three OTUs) species. During the cheesemaking process, the total abundance of LAB-related OTUs increased about 10-fold (from 2.2 to 25.8% of the total OTUs). In particular, OTU 145 and 155, affiliated with *Lactobacillus* sp. and *Leuconostoc mesenteroides*, increased about 12- and 23-fold, respectively (Appendix A). The total abundance of bacteria belonging to the O category (OTUs 23, 122 and 285) decreased about 3.4-fold from 1.44 to 0.42% of the total OTUs during the cheesemaking process.

## 4. Discussion

The main aim of this study was to assess the effect of the diversity of the milk microbiome of two different sheep breeds, Comisana and Lacaune, on the microbial community of artisanal Pecorino Romano-like cheese. The analysis was carried out on samples collected at various stages of the cheesemaking process (milk, curd and Pecorino-like cheese) using high-throughput sequencing of the 16S rRNA gene.

The rarefaction curves reported in Figure 1 indicate that the 16S rRNA datasets represent the bacterial community’s complexity well.

To gain information about the fingerprints of the microbiomes of Comisana and Lacaune milk, we combined OTU data from replicate samples. We used the resulting datasets for Venn analysis and PCA. The comparative analysis datasets, through the Venn diagram, allowed us to identify common (shared taxa) and unique (accessory taxa) OTUs occurring in the two milk samples. Principal component analysis allowed us to identify the OTUs whose abundance profiles varied between the two microbiome datasets. A similar approach was used to evaluate the contribution of the milk and curd microbiome to the cheese microbial community. The Venn diagram (Figure 2, Panel A) showed that the Comisana and Lacaune milk datasets shared 25 OTUs, representing about 97.5% and 96.1% of the total reads, respectively (Appendix A).

PCA indicated that the Comisana and Lacaune milk microbiomes were markedly distinct and that differences in the two datasets were due to the relative abundance of the shared taxa (Figure 3). In both milk microbiomes, 44 and 59% of the total reads were associated with the *P. veronii*-affiliated OTU 340, representing more than 97% of the total reads belonging to *Pseudomonadaceae*. These results on the occurrence of specific taxa in samples collected from farms of the same geographic area support the hypothesis that the environment shapes the milk microbiota and impacts the microbial terroir of artisanal fermented products [36].

*Pseudomonas veronii* is a non-pathogenic environmental microorganism originally isolated from mineral water. It is known for its ability to degrade aromatic compounds [37,38,39]. The presence of *P. veronii* in the milk microbiome has already been reported for buffalo and other mammals [29,40,41] but has yet to be observed in the milk microbiome of different sheep breeds, such as Assaf dairy ewes [42].

PCA also revealed differences in the abundance of OTUs affiliated with *Xanthomonadaceae* (OTU 348 and 350). In CSB milk samples, these OTUs were about 2-fold higher than those in the LSB milk datasets, indicating that the breed and farming environments potentially influence the presence of these taxa in the milk microbiome. The related taxa belonged to *Stenotrophomonas*, a genus whose members are known components of the core milk microbiome of goats [43] and cows [44]. *Stenotrophomonas* comprises psychrotrophic and proteolytic strains, which can be involved in bovine mastitis [45] and raw milk spoilage [46]. Notably, OTUs 348 and 350 were not affiliated with *Stenotrophomonas maltophilia*, a pathogen reported to be associated with human respiratory infections [47,48,49].

Another significant difference between the two milk microbiomes was the abundance of the *Enterobacteriaceae* and *Streptococcaceae* families and their representative OTUs (IDs 307, 315 and 159; Figure 3). OTU 307 represented more than 98% of the total *Enterobacteriaceae*-affiliated reads in both milk microbiomes, whereas *Enterobacteriaceae*-affiliated OTU 315 was detected only in the LSB milk datasets. OTU 159 represented more than 97% of the total reads belonging to *Streptococcaceae* in the CSB milk samples and only 4% of the *Streptococcaceae*-affiliated reads in the LSB microbiome. The differential abundance of OTUs of the shared taxa underlines the effect of the dairy farming practice on the composition of the milk microbial community.

The shared OTUs of the Comisana and Lacaune microbiomes reported in this work show differences with the Assaf dairy ewe microbiome described by Esteban-Blanco et al. [42]. These authors reported that the milk microbiome of healthy Assaf sheep comprised five dominant genera: *Corynebacterium*, *Escherichia/Shigella*, *Lactobacillus*, *Staphylococcus* and *Streptococcus* [42]. In contrast, we identified 22 different genera that were shared in the microbiomes of Comisana and Lacaune sheep. Three of them, *Lactobacillus*, *Streptococcus* and *Corynebacterium*, occurred in the microbiomes of all three sheep breeds. We detected the presence of an OTU affiliated with *Enterobacteriaceae* but not belonging to the *Escherichia*/*Shigella* phylogroup. *Staphylococcus*-affiliated OTUs were observed in the Assaf and Lacaune milk microbiomes but were absent in Comisana milk. These data suggest that the *Staphylococcus* genus is not part of the core genera of the sheep milk microbiome.

Principal component analysis of different datasets revealed that the Comisana and Lacaune raw milk coagulation curds possessed a distinct complex microbiome (Figure 4). As shown in Appendix A, the total OTUs significantly increased from the milk to the curd in both samples, indicating that the combination of rennet and cheese starter used in the two cheesemaking processes, as expected, significantly affected the biodiversity of the curd microbiome. In the CSB datasets, the number of OTUs in the curd samples was 4.1-fold higher than that in the corresponding milk (207 vs. 50 OTUs; Appendix A), whereas in the LSB datasets, this number increased about 2.5-fold (265 vs. 66; Appendix A). About 46 (23 out of 50 OTUs) and 65% (43 out of 66 OTUs) of the total OTUs occurring in CSB and LSB milk were not present in the corresponding curd (Figure 5). The comparative analysis of the milk and curd microbiomes revealed that the *Xanthomonadaceae*-associated OTUs (IDs 347, 348 and 350) of the shared milk taxa were drastically reduced or disappeared after thermal treatment (Appendix A). The same analysis also revealed that the most representative OTUs of the shared milk taxa affiliated with *Pseudomonodaceae* (ID 340) and *Enterobacteriaceae* (ID 307) disappeared during the cheesemaking process. In contrast, OTU ID 159 (*Streptococcoccus* sp.) decreased 25-fold in CSB curd and increased about 8.2-fold in LSB curd compared to the corresponding milk samples.

The data reported in Figure 5, Panel A, also indicate the presence of 44 OTUs shared only between the two curd datasets, whose presence could have been due to environmental contamination. The relative abundance of most of these shared OTUs was below 0.1% (36 OTUs in the CSB curd dataset and 30 OTUs in the LSB dataset), and only a few of them (1 OTU in the CSB curd dataset and 6 OTUs in the LSB dataset) were present in the corresponding mid-ripened cheese at a relative abundance higher than 0.1% (Appendix A).

The PCA of the curd datasets, which explains over 99% of the total variance (Figure 5, Panel B), indicated that the significant differences between the two curd microbiomes were due to the abundance of eight shared OTUs and two OTUs (ID 79 and 271) that were present only in the LSB samples. The latter were identified as *F. frigidarium* (OTU 79) and *Comamonas* sp. (OTU 271), two environmental taxa whose presence was reported in artisan Mongolian sheep cheese by Guo et al. [50].

Two out of the eight shared OTUs belonged to the *Serratia* (ID 315) and *Pseudomonas* (ID 337) genera, and their presence could be related to environmental contamination, since these microorganisms are ubiquitous in water, soil and other environments [51,52,53]. Both genera include species involved in food spoilage often associated with dairy products that are recognized as resident microbiota of food processing plants for their ability to produce biofilms resistant to cleaning procedures [22,54,55,56].

Furthermore, Ruta et al. [57] reported the presence of *Serratia* and *Pseudomonas* in Pecorino Siciliano curd samples collected from five different farms. In both cheese ripening processes, the abundance of *P. fragi*-associated OTU 337 significantly decreased (25-fold in CSB samples) or disappeared (in LSB samples; Appendix A). This effect can be related to the environmental changes associated with Pecorino-like cheese production (high salinity and low pH), which inhibit this taxon’s growth and survival [58,59]. Comparing the microbiome pattern of curd and the corresponding cheese, we observed a different trend in the abundance of *Serratia*-associated OTU 315. In CSB samples, this increased about 8.5-fold from the curd to the cheese, whereas in LSB samples, its abundance decreased by up to 0.03% of the total OTUs. Members of the *Serratia* genus are commonly isolated from cheese. Todaro et al. [60], analyzing the effect of salting technologies on the cheese microbiome, reported the presence of *Serratia* in different PDO Pecorino cheeses. These authors suggested that the survival of unwanted bacteria, including *Serratia*, is inversely correlated with the abundance of LAB. Our data indicate that *Lactobacillales*-affiliated OTUs represented more than 92% of the total OTUs in LSB cheese samples, in which we observed a low level of *Serratia*. Moreover, *Serratia* represented about one-third of the total cheese microbiome in CSB cheese samples, in which *Lactobacillales* were only 58% of the total OTUs. Both *Lactobacillales* and *Serratia* are known to produce bacteriocins active against Gram-negative bacteria, including *Escherichia coli* and *Pseudomonas* [61,62,63,64,65].

Moreover, bacteriocins produced by LAB can be active against *Serratia*, which can be valuable in the cheesemaking sector for reducing the development of these unwanted spoilage microorganisms. A more detailed analysis of the *Lactobacilalles*-affiliated OTUs indicated a strong effect of the cheesemaking process on the number and abundance of these taxa. No OTU related to *Carnobacteriaceae* and *Enterococcaceae* was present in CSB cheese samples, whereas in the LSB cheese samples, they represented about 0.43% and 0.81% of the entire microbiome, respectively. Members of the *Streptococcus* (St) and *Lactobacillus* (Lb) genera were differentially represented in the two cheeses. Taxa belonging to these genera were more abundant in LSB (3.80%, St; 8.90%, Lb) cheese samples than those in CSB (0.02%, St; 0.3, Lb) cheese samples.

The comparative analysis of the two Pecorino cheese microbiomes revealed that the main differences were related to five OTUs: three LAB-affiliated OTUs (*Lactobacillus* sp. OTU 145, *L. mesenteroides* OTU 155 and *Lactococcus* sp. OTU 156) and two environmental contaminants (*Serratia* sp. OTU 315 and *P. fragi* OTU 337). The relative abundance of these taxa was 96 (CSB) and 85% (LSB) of the total OTUs, respectively (Appendix A). Only two were present in milk and the corresponding cheese at a detectable level (OTU ID 155 and 156 in CSB samples; OTU 156 and 315 in LSB samples). Interestingly, OTU 156, corresponding to the LAB involved in the acidification process, was 10-fold more abundant in LSB (1.02% of the total OTUs) milk samples than that in CSB (0.1% of the total OTUs) milk samples. Despite the data reported in Appendix A indicating that the growth rates of *Lactococcus*-affiliated OTU 156, from milk to curd, were similar in the two datasets, the different initial concentrations of this taxa in the raw CSB and LSB milk affected the acidification process, generating environmental conditions that, in the LSB samples, favored the development of natural non-starter lactic acid bacteria (NSLAB; *L. mesenteroides* affiliated OTU 155) and the containment of *Serratia* and *Pseudomonas* contaminants.

These data indicate that the structure and composition of the Lacaune sheep breed microbiota are valuable in the artisanal process to obtain Pecorino-like cheese with a higher concentration of NSLAB (*L. mesenteroides*), which can have a positive effect on flavor development, and a lower concentration of spoilage bacteria (*Serratia* sp. and *P. fragi*).

The presence of unique OTUs in both cheese samples can be related to taxa (e.g., *Lactobacillales* and *Clostridiales*) whose relative abundance falls below the detectable limit in the milk and curd microbiomes. Based on our results, establishing the origin of these taxa (milk or curd) is impossible. However, it is worth mentioning that, taken together, they represent only a minor part of the entire cheese microbiome: 1.5% in LSB and 1.7% in CSB (Appendix A).

## 5. Conclusions

In conclusion, our study revealed differences in the milk microbiomes of Comisana and Lacaune sheep. These differences were associated with the relative abundance of starter and non-starter LAB, which were important for the organoleptic and safety properties of artisanal cheeses. The profiling of the Comisana and Lacaune milk microbiomes allowed us to determine the effect of the environment on the milk microbiomes, identify the set of the core genera that form the sheep milk microbiome and analyze the temporal changes in the shared OTU abundance occurring during cheesemaking. Our data underline the importance of Next-Generation Sequencing as a valuable tool for developing a fermentation process that valorizes the autochthonous microbiota and increases the safety of artisanal products. In the future, it will be interesting to match the NGS analysis with sensory evaluation studies to correlate the microbial evolution during the cheesemaking process with the sensory profile of artisanal Pecorino-like cheeses.

## Figures and Tables

**Figure 1 foods-13-01728-f001:**
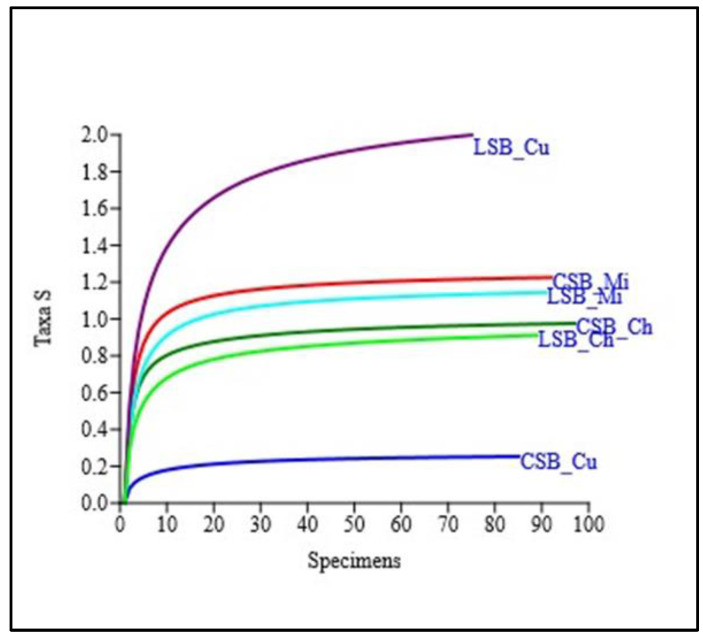
Rarefaction curves for the samples of the five different farms. LSB = Lacaune Sheep Breed, CSB = Comisana Sheep Breed, Mi = milk samples, Cu = curd samples, Ch = cheese samples.

**Figure 2 foods-13-01728-f002:**
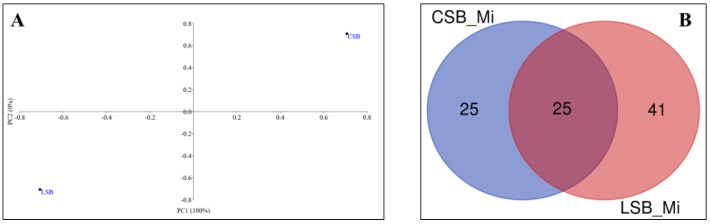
Comparative analysis of milk microbiomes. (**A**) Score plot showing principal components (PCO) 1 and 2 calculated with the abundance matrix of the CSB and LSB milk microbiota. Each symbol represents one microbiota. The plot model explains 100% of the total data variance. (**B**) Venn diagram showing the number of shared and unique OTUs (≥2 reads) in the CSB and LSB milk microbiotas.

**Figure 3 foods-13-01728-f003:**
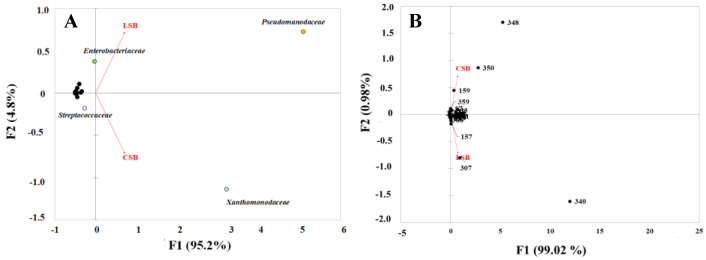
Principal component analysis (PCA) of taxa occurring in CSB and LSB milk microbiotas based on the relative abundance distribution at the family (**A**) and OTU (**B**) levels.

**Figure 4 foods-13-01728-f004:**
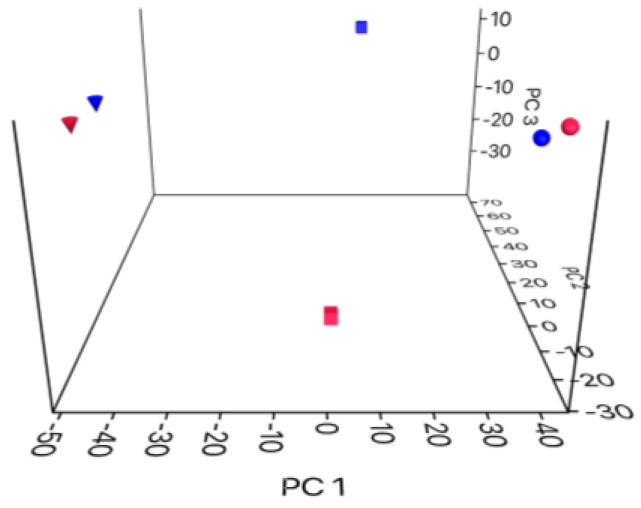
Principal component analysis of taxa occurring in the CSB (blue symbols) and LSB (red symbols) in milk (circle), curd (square) and cheese (inverted triangle) microbiotas based on the relative abundance distribution at the family level.

**Figure 5 foods-13-01728-f005:**
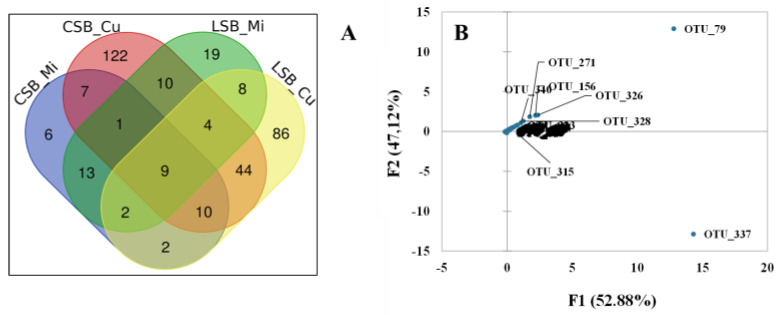
Comparative analysis of curd microbiomes. (**A**) The Venn diagram shows the number of shared and unique OTUs (≥2 reads) in the CSB and LSB milk and curd microbiotas. (**B**) Principal component analysis (PCA) of taxa occurring in the CSB and LSB curd microbiotas is based on the relative abundance distribution at the OTU level.

**Figure 6 foods-13-01728-f006:**
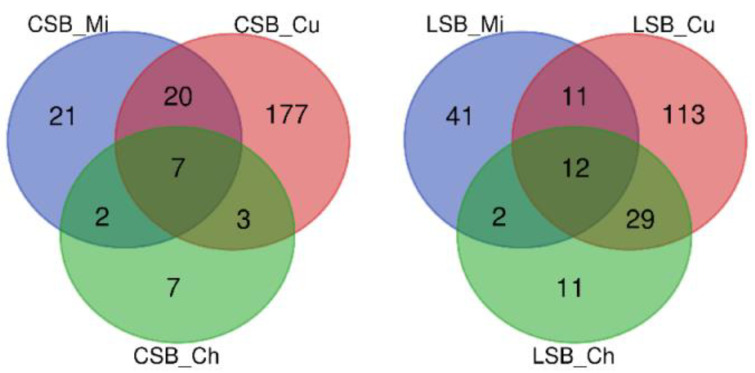
The Venn diagrams show the number of shared and unique OTUs (≥2 reads) in the milk, curd and cheese microbiotas from the CSB (Left Panel) and LSB (Right Panel) cheesemaking processes.

## Data Availability

The original contributions presented in the study are included in the article/Appendix A, further inquiries can be directed to the corresponding author.

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
