# Peer review of "Microbial Evolution in Artisanal Pecorino-like Cheeses Produced from Two Farms Managing Two Different Breeds of Sheep (Comisana and Lacaune)"

_foods, 2024, doi:10.3390/foods13111728_

Round 1

Reviewer 1 Report

Comments and Suggestions for Authors

Overview

The work deals with monitoring the evolution of the microbiome composition from milk to middle-aged (10-20 days) Pecorino-like cheese, a semi-hard cheese type obtained with raw or heat-treated sheep milk. Microbiota monitoring was done using a high throughput method of 16S rRNA gene sequencing in two artisanal productions that used milk from two different sheep breeds (Comisana and Lacaune). The comparative analysis of the bacterial community composition revealed significant differences in the presence and abundance of specific taxa in the milk microbiome of the two sheep breeds. Next-Generation Sequencing (NGS) analysis also revealed differences in the curd microbiome which were related to dairy farming practices and had a relevant effect on the final constitution of the Pecorino-like cheese microbiome. The authors conclude that NGS is a valuable tool to develop a fermentation process that valorizes the autochthonous microbiota.

In my opinion, the work has some scientific interest and is generally well structured. The experimental design and models applied for data analysis also seem correct and appropriate. However, there are some conceptual errors in relation to the present study that should be clarified. The article compares two artisanal productions of raw milk and special emphasis is placed on the breed of sheep as a possible variation factor in the cheeses obtained in both productions. When the results related to the microbiota present in the two sheep’s milk are shown, the expression “milk core microbiome” is used. In relation to this assessment, it is worth remembering that a healthy mammary gland does not provide any type of microorganisms to the milk, and this raw material is considered to be theoretically sterile inside a gland that is not affected by mastitis. It is common that some (non-pathogenic or non-infectious) microorganisms can access the interior of the udder through the teat canal, but this would be defined as environmental contamination rather than “milk core microbiome”. On the other hand, it is inevitable that milk becomes contaminated in the canal and on the surface of the teat as it exits the udder, but this contamination also comes from the environment. In fact, the differences in microbiological quality of cow’s milk and sheep’s milk (the latter generally much more contaminated) are basically due to the fact that the hygienic conditions in the handling and milking of sheep are not usually as strict as in the case of cattle. I believe that these aspects should be taken into account when reviewing the sections of the manuscript that refer to the microbiota of sheep’s milk. A difference in the microbial communities of the milk from both productions should not be justified exclusively as a consequence of the “breed factor”, but rather due to different environmental contaminations. On another matter, I think that the article should make a brief description of the operations carried out in the cheese making process that could have a relevant impact on microbial growth and selection of the microbiota (milk cooling, heat treatment, possible use of starter cultures obtained from whey or fermented milk, temperature and relative humidity in the ripening room). This information could help explain or discuss some of the results. Finally, the authors do not make any mention of the sensory characteristics of the cheeses obtained in the two productions, nor do they even comment on whether their sensory quality is different. Assuming that differences were observed between both batches of cheese and these differences were related to the abundance of some OTUs, perhaps it could be suggested that NGS could contribute to “valorizing the autochthonous (adventitious or contaminating) microbiota”.

From a formal point of view, the manuscript is generally well written, although I have found some grammatical and typographical errors that I mention in the detailed comments.

General comments

- Abbreviations must be previously defined (in the abstract, the main text, and the first figure or table) before being used, the first time in parentheses.

- The commercial data of the suppliers of materials and reagents must be provided in their entirety (city, country) the first time they are mentioned.

- Avoid starting a sentence with an abbreviation or an abbreviated word such as the initial of a genus name.

- The names of the bacterial genera and species should be written in italics.

Detailed comments

Title

- Change to e.g.: “…and microbial evolution during artisanal Pecorino-like cheese making”.

Abstract

- Line 15. Change to “Pecorino-like cheese”.

- Line 18. Change to “Next-Generation Sequencing”. Acronyms/abbreviations should be defined the first time they appear in each of three sections: the abstract; the main text; the first figure or table.

Introduction

- Lines 34–35. Why is milk heated to temperatures of 45-48 ºC? What is the duration of this treatment? Minimum thermization treatments are usually carried out at 62ºC for 10-20 s, and and minimum (HTST) pasteurization treatments are performed at 72 ºC for 15 s. Write “sheep’s” with the correct apostrophe symbol (also on line 41, “milk’s”).

- Line 45. Change “volatile acids” to “volatile compounds”.

- Line 58. The reference [26] is about “bacterial spoilers in beefsteaks”.

- Line 63. Scroll the paragraph up.

Materials and Methods

- Lines 71–72. Where do “native starter cultures” come from? Are they obtained from whey or raw fermented milk? Are these cheeses made with spontaneous fermentations caused by microorganisms present in the milk? If whey cultures or fermented or acidified raw milk are used, take this factor into account when commenting on the results regarding the “curd microbiome”.

- Line 72. “Pecorino Romano-like” or “Pecorino-like” as in the article title?

- Line 73. “10-20 days” seems like a rather wide ripening interval for a semi-hard cheese.

- Line 76. How was the fat content of milk reduced? Was the milk centrifuged?

- Line 77–78. What were the centrifugation conditions for partially skimmed milk to separate the cellular contents? Change to “–20ºC” (keep the negative symbol and the number together).

- Line 81. Provide the commercial data (city, country) for “Qiagen” (“Qiagen GmbH, Hilden, Germany”).

- Line 84. Always express centrifugation units as relative centrifugal force (“× g”). Otherwise, the operation could not be reproduced in a different centrifuge.

- Line 86. Provide the units for “10,000” (relative centrifugal force).

- Lines 100–101. Change to “(Qiagen GmbH)”.

- Line 109. Add “, USA” after “WA”.

Results

- Line 118. Change to “Next-Generation Sequencing”. Avoid starting a sentence with an abbreviation or an abbreviated word.

- Lines 148–150. The families Pseudomonadaceae and Xanthomonadaceae contain genera and species that show psychrotrophic growth. Was the milk subjected to cooling before cheese making?

- Lines 163–164. Enterobacteriaceae were present around 8 times higher in LSB milk than in CSB. Should this fact be related to poorer hygienic conditions in milking and obtaining LSB milk? Enterobacteriaceae from manure, soil and water often contaminate milk on the outside of the udder.

- Line 191. Change to “93.4% (CSB)”.

- Lines 218–221. The OTUs shared by milk and cheese affiliated with their most abundant taxa in milk and cheese (?). The sentence is somewhat confusing.

- Lines 254–258. Write all names of microbial families, genera, and species in cursive letters. Change to “Rothia”, “Pasteurelleae”, “Pseudoalteromonadaceae”, “Lactobacillus paralimentarius”, “Lactococcus garvieae”, “Carnobacteriaceae”. Extreme care should be taken both in checking the current taxonomy and in spelling the names of the microorganisms correctly.

Discussion

- Line 300. “Pecorino-like cheese” or “Pecorino Romano-like cheese”? Be consistent and choose one of the two names for the entire text and the title.

- Line 309. Change to “Principal component analysis”. Avoid starting a sentence with an abbreviation.

- Lines 317–321. The prevalence of Pseudomonadaceae (psychrotrophic bacteria usually carried in the washing water) in raw milk could be related to the cooling and cold maintenance of the milk. Milk in a healthy udder is assumed to be sterile, so it is foreseeable that the environment and environmental contamination configure the microbiota of raw milk.

- Line 322. Change to “Pseudomonas veronii”. Avoid starting a sentence with an abbreviated word such as the initial of a genus name.

- Line 326. Add a period after “[47]”.

- Line 327. Change to “The PCA also revealed…”

- Line 330. Change to “The related taxa”.

- Line 355. Staphylococci could contaminate milk from the skin of the udder or as a result of handling during milking.

- Line 367. What were the conditions of the heat treatment of milk? Was the milk simply heated to temperatures of 45-48ºC? These temperatures would not cause the destruction of psychrotrophic bacteria or enterobacteria. It would be very interesting to describe the operations carried out in the cheese making process that could have a relevant impact on microbial growth and selection of the microbiota.

- Line 381. Delete “two”.

- Line 396. Could DNA from bacterial cells that were no longer viable be amplified and and consequently sequenced?

- Line 418. Change to “than in CSB”.

- Lines 435–438. In my opinion, the sentence is meaningless. As far as I know, the milk inside a healthy udder (not clinically or subclinically infected) is a sterile material, which begins to become contaminated as it exits through the teat canal and on its surface. The differences in the microbial groups found in raw milk should be attributed mainly to external or environmental contamination, and it should not be assumed that each sheep breed produces a characteristic milk with a specific microbiota.

- Lines 439–441. Microorganisms can also contaminate milk during cheese making, unless the process takes place under aseptic conditions.

Conclusions

- Lines 446–456. Join the paragraphs, do not use full stops.

- Lines 452–453. How was “the impact of changes in core OTU abundance on cheese making” evaluated?

- Lines 454–456. Why do you say that NGS is a valuable tool to develop a “fermentation process that valorizes the autochthonous microbiota”? What were the sensory characteristics or sensory quality of the cheeses obtained in the two productions?

References

- pp. 11–14. The number of references (70) seems excessive. If possible, remove 5-10 references.

Author Response

Responses to Reviewer 1’s comments

General comments

Comment: Abbreviations must be previously defined (in the abstract, the main text, and the first figure or table) before being used, the first time in parentheses.

Response: the manuscript has been revised as suggested by the reviewer

Comment: The commercial data of the suppliers of materials and reagents must be provided in their entirety (city, country) the first time they are mentioned.

Response: The suggestion has been accepted

Comment: Avoid starting a sentence with an abbreviation or an abbreviated word such as the initial of a genus name.

Response: The suggestion has been accepted

Comment: The names of the bacterial genera and species should be written in italics.

Response: the text has been revised as suggested by the reviewer and all bacterial genera or species has been written in italics

Comment: Title - Change to e.g.: “…and microbial evolution during artisanal Pecorino-like cheese making”.

Response: The title was changed as suggested

Comment: Line 15. Change to “Pecorino-like cheese”.

Response: the sentence has been changed as suggested (Line 15 in the revised manuscript)

Comment: Line 18. Change to “Next-Generation Sequencing”. Acronyms/abbreviations should be defined the first time they appear in each of three sections: the abstract; the main text; the first figure or table.

Response: the text has been revised as suggested by the reviewer (Line 18 in the revised manuscript)

Comment: Lines 34–35. Why is milk heated to temperatures of 45-48ºC? What is the duration of this treatment? Minimum thermization treatments are usually carried out at 62ºC for 10-20 s, and minimum (HTST) pasteurization treatments are performed at 72ºC for 15 s. Write “sheep’s” with the correct apostrophe symbol (also on line 41, “milk’s”).

Response: the sentence was changed to provide a more precise description of the cheesemaking process “sheep's milk by traditional procedures [10]. Refrigerated milk is filtered, heat-treated (62-68°C for at least 15 seconds), and inoculated with native milk ferment cultures. The milk curdles between 38°C and 40°C with adding lamb’s rennet crust. The cooking of the soured milk is performed at a temperature between 45°C-48°C [11]” (Lines 34-38 in the revised manuscript).

Comment: Line 45. Change “volatile acids” to “volatile compounds”.

Response: the text has been revised as suggested by the reviewer (Line 60 in the revised manuscript)

Comment: Line 58. The reference [26] is about “bacterial spoilers in beefsteaks”.

Response: The author thanks the reviewer for this comment, and the reference to De Filippis's work has been changed to “De Filippis, F.; Parente, E.; Ercolini, D. Metagenomics insights into food fermentations. Microb. Biotechnol. 2017, 10, 91-102.”

Comment: Line 63. Scroll the paragraph up.

Response: The suggestion has been accepted

Comment: Lines 71–72. Where do “native starter cultures” come from? Are they obtained from whey or raw fermented milk? Are these cheeses made with spontaneous fermentations caused by microorganisms present in the milk? If whey cultures or fermented or acidified raw milk are used, take this factor into account when commenting on the results regarding the “curd microbiome”.

Response: the nature of the starter culture was explained (scotta-innesto) (Lines 86-87 in the revised manuscript)

Comment: Line 72. “Pecorino Romano-like” or “Pecorino-like” as in the article title?

Response: The term Pecorino-like cheese was used

Comment: Line 73. “10-20 days” seems like a rather wide ripening interval for a semi-hard cheese.

Response: the age of ripening was corrected (Line 89 in the revised manuscript)

Comment: Line 76. How was the fat content of milk reduced? Was the milk centrifuged?

Response: by centrifugation. The text has been revised to clarify the procedure (Lines 91-93 in the revised manuscript).

Comment: Line 77–78. What were the centrifugation conditions for partially skimmed milk to separate the cellular contents? Change to “–20ºC” (keep the negative symbol and the number together).

Response: The text has been revised to clarify the procedure (Lines 91-95 in the revised manuscript).

Comment: Line 81. Provide the commercial data (city, country) for “Qiagen” (“Qiagen GmbH, Hilden, Germany”).

Response: The suggestion has been accepted (Line 98 in the revised manuscript)

Comment: Line 84. Always express centrifugation units as relative centrifugal force (“×g”). Otherwise, the operation could not be reproduced in a different centrifuge.

Response: The centrifugation unit was changed

Comment: Line 86. Provide the units for “10,000” (relative centrifugal force).

Response: the centrifugation unit was added (Line 104 in the revised manuscript)

Comment: Lines 100–101. Change to “(Qiagen GmbH)”.

Response: The suggestion has been accepted (Line 159 in the revised manuscript)

Comment: Line 109. Add “, USA” after “WA”.

Response: The suggestion has been accepted (Line 168 in the revised manuscript)

Comment: Line 118. Change to “Next-Generation Sequencing”. Avoid starting a sentence with an abbreviation or an abbreviated word.

Response: The suggestion has been accepted (Line 175 in the revised manuscript)

Comment: Lines 148–150. The families Pseudomonadaceae and Xanthomonadaceae contain genera and species that show psychrotrophic growth. Was the milk subjected to cooling before cheese making?

Response: Under the PDO regulation, the milk is cooled before cheesemaking, which could favor the growth of psychrotrophic bacteria. However, the data reported in Table S3 indicated that the abundance of the specific OTUs belonging to these families varied in function of the milk breed, indicating that the population of the psychrotrophic bacteria is different.

Comment: Lines 163–164. Enterobacteriaceae were present around 8 times higher in LSB milk than in CSB. Should this fact be related to poorer hygienic conditions in milking and obtaining LSB milk? Enterobacteriaceae from manure, soil and water often contaminate milk on the outside of the udder.

Response: yes. The presence of Enterobacteriaceae may be due to the hygienic conditions, which are part of what we defined as “dairy farming practices” (Lines 19 and 420 in the revised manuscript).

Comment: Line 191. Change to “93.4% (CSB)”.

Response: The suggestion has been accepted (Line 257 in the revised manuscript)

Comment: Lines 218–221. The OTUs shared by milk and cheese affiliated with their most abundant taxa in milk and cheese (?). The sentence is somewhat confusing.

Response: the text has been revised as follows: “included, together with minor taxa, the Pseudomonas veronii-affiliated OTU 340 (the most abundant OTU in the milk dataset) and Lactococcus sp.-affiliated OTU 156 (the most abundant taxon in cheese datasets; Table S5; Lines 286-288 in the revised manuscript)”.

Comment: Write all names of microbial families, genera, and species in cursive letters. Change to “Rothia”, “Pasteurelleae”, “Pseudoalteromonadaceae”, “Lactobacillus paralimentarius”, “Lactococcus garvieae”, “Carnobacteriaceae”. Extreme care should be taken both in checking the current taxonomy and in spelling the names of the microorganisms correctly.

Response: The suggestion has been accepted (Lines 329-332 in the revised manuscript)

Comment: Line 300. “Pecorino-like cheese” or “Pecorino Romano-like cheese”? Be consistent and choose one of the two names for the entire text and the title.

Response: The term Pecorino-like cheese was used (Line 375 in the revised manuscript)

Comment: Line 309. Change to “Principal component analysis”. Avoid starting a sentence with an abbreviation.

Response: The suggestion has been accepted (Line 383 in the revised manuscript)

Comment: Lines 317–321. The prevalence of Pseudomonadaceae (psychrotrophic bacteria usually carried in the washing water) in raw milk could be related to the cooling and cold maintenance of the milk. Milk in a healthy udder is assumed to be sterile, so it is foreseeable that the environment and environmental contamination configure the microbiota of raw milk.

Response: our data on Pseudomonadaceae indicated that the abundance of this family is associated with one OTU (P. veronii- affiliated OTU 340). The Reviewer agrees with our conclusion that “the occurrence of specific taxa in samples collected from farms of the same geographic area supported the hypothesis that the environment shapes the milk microbiota.”

Comment: Line 322. Change to “Pseudomonas veronii”. Avoid starting a sentence with an abbreviated word such as the initial of a genus name.

Response: The suggestion has been accepted (Line 399 in the revised manuscript)

Comment: Line 326. Add a period after “[47]”.

Response: The suggestion has been accepted (Line 403 in the revised manuscript)

Comment: Line 327. Change to “The PCA also revealed…”

Response: The suggestion has been accepted (Line 404 in the revised manuscript)

Comment: Line 330. Change to “The related taxa”.

Response: The suggestion has been accepted (Line 407 in the revised manuscript)

Comment: Line 355. Staphylococci could contaminate milk from the skin of the udder or as a result of handling during milking.

Response: we agree with the reviewer that the conclusion by Esteban-Blanco on the association of the Staphylococcus genus on the milk core microbiome of the Assaf sheep breed is questionable. As reported in our manuscript, “Staphylococcus genus is not part of the core microbiome of the sheep milk” (Line 432 in the revised manuscript)

Comment: Line 367. What were the conditions of the heat treatment of milk? Was the milk simply heated to temperatures of 45-48ºC? These temperatures would not cause the destruction of psychrotrophic bacteria or enterobacteria. It would be very interesting to describe the operations carried out in the cheese making process that could have a relevant impact on microbial growth and selection of the microbiota.

Response: see the response to comment Lines 71–72. A description of the process has been included in the revised manuscript.

Comment: Line 381. Delete “two”.

Response: The suggestion has been accepted (Line 463 in the revised manuscript)

Comment: Line 396. Could DNA from bacterial cells that were no longer viable be amplified and consequently sequenced?

Response: The protocol we have used allows us to recover DNA from all intact cells (viable and non-viable).

Comment: Line 418. Change to “than in CSB”.

Response: The suggestion has been accepted (Line 501 in the revised manuscript)

Comment: Lines 435–438. In my opinion, the sentence is meaningless. As far as I know, the milk inside a healthy udder (not clinically or subclinically infected) is a sterile material, which begins to become contaminated as it exits through the teat canal and on its surface. The differences in the microbial groups found in raw milk should be attributed mainly to external or environmental contamination, and it should not be assumed that each sheep breed produces a characteristic milk with a specific microbiota.

Response: when secreted from a healthy udder the milk is almost sterile. The topic has been discussed in several papers including Pascal Rainard (2017; doi: 10.1186/s13567-017-0429-2) and Derakhshani et al. (2018; doi: 10.3168/jds.2018-14860)

Comment: Lines 439–441. Microorganisms can also contaminate milk during cheese making, unless the process takes place under aseptic conditions.

Response: One of the main goals of this work was to evaluate the effect of the cheesemaking process on the microbiome of fermented foods. The data presented in this manuscript demonstrated how the population of the microbial contaminants introduced with the curd evolves during the process.

Comment: Lines 446–456. Join the paragraphs, do not use full stops.

Response: The suggestion has been accepted

Comment: Lines 452–453. How was “the impact of changes in core OTU abundance on cheese making” evaluated?

Response: We modified the test to “analyze the temporal changes in the core OTU abundance occurring during the cheesemaking” to clarify the conclusion. (Line 715 in the revised manuscript)

Comment: Lines 454–456. Why do you say that NGS is a valuable tool to develop a “fermentation process that valorizes the autochthonous microbiota”? What were the sensory characteristics or sensory quality of the cheeses obtained in the two productions?

Response: Knowing the milk microbiome can provide valuable hints to improve the fermentation process for stimulating the growth of specific taxa whose metabolism leads to the production of compounds enhancing the cheese sensory characteristics.

Comment: pp. 11–14. The number of references (70) seems excessive. If possible, remove 5-10 references.

Response: The suggestion has been accepted, and five references were removed from the manuscript.

Reviewer 2 Report

Comments and Suggestions for Authors

This manuscript describes how the microorganisms in raw sheep milk change in type and numbers as the milk is converted into curd and then cheese similar to Pecorino. The paper goes into detail on the species of bacteria and their population using data obtained by 16S rRNA gene sequencing. The descriptions and figures illustrating the main points were clear and informative. Some comments:

Title: The microbial evolution can't be during cheese, but during some process. Change to "during manufacture of artisanal Pecorino-like cheese" or "during artisanal Pecorino-like cheese manufacture".

Line 73: 50 g seems to be a small sample. Add a sentence explaining why you felt that size was adequate.

Line 85: Not sure what "amended" means here.

Lines 115-117: This sentence is an explanation of why you did the work, and belongs in the introduction. Combine it with lines 63-66. 

Line 148: Should use the plural form, phyla.

Lines 319-321: You can expand a little on the fact that environment affects the microbiota. It is why many people like artisanal cheese better than the mass-produced product.

Lines 359-360: The starter and rennet would obviously affect the microbiome in the curd, so "as expected" should go in this sentence.

Lines 408-412: Bacteriocins from LAB are active against Serratia, but bacteriocins from Serratia are active against E. coli and Pseudomonas. Does LAB cancel out the benefit of Serratia bacteriocins?

Author Response

Responses to Reviewer 2’s comments

 Comment: The microbial evolution can't be during cheese, but during some process. Change to "during manufacture of artisanal Pecorino-like cheese" or "during artisanal Pecorino-like cheese manufacture".

Response: Thanks for the suggestion; we changed the tile to read “Microbiome diversity in raw milk from Comisana and Lacaune sheep and microbial evolution during artisanal Pecorino-like cheese making.”

Comment: Line 73: 50 g seems to be a small sample. Add a sentence explaining why you felt that size was adequate.

Response: the text has been revised as follows: “a representative amount of curd (50 g) and middle-aged (20 days) Pecorino-like cheese (1/5 of a 250 g cheese form) (Lines 88-89 in the revised manuscript).

Line 85: Not sure what "amended" means here.

Response: the suggestion has been accepted (Line 102 in the revised manuscript)

Lines 115-117: This sentence is an explanation of why you did the work, and belongs in the introduction. Combine it with lines 63-66.

Response: the suggestion has been accepted (Lines 77-80 in the revised manuscript)

Line 148: Should use the plural form, phyla.

Response: the suggestion has been accepted (Line 213 in the revised manuscript)

Lines 319-321: You can expand a little on the fact that environment affects the microbiota. It is why many people like artisanal cheese better than the mass-produced product.

Response: the suggestion has been accepted, and the text has been modified as follows: “and impact the microbial terroir of artisanal fermented products” (Lines 397-398 in the revised manuscript).

Lines 359-360: The starter and rennet would obviously affect the microbiome in the curd, so "as expected" should go in this sentence.

Response: the suggestion has been accepted (Line 436 in the revised manuscript)

Lines 408-412: Bacteriocins from LAB are active against Serratia, but bacteriocins from Serratia are active against E. coli and Pseudomonas. Does LAB cancel out the benefit of Serratia bacteriocins?

Response: Serratia is known to be a microorganism responsible for cheese alterations; a reduction of Serratia’s population is valuable in a cheesemaking process

Round 2

Reviewer 1 Report

Comments and Suggestions for Authors

Overview

The authors have responded point by point to all my questions and observations. The authors have also taken into consideration most of my comments when drafting the revised version of the manuscript and have clarified some of the doubts I had regarding the study.

However, the critical point of this study continues to be that concerning the attribution of the differences in the microbiota of the raw milk sampled on two different farms to the different breeds of sheep that are managed on the farms, and the use of the expression “milk core microbiome”. In my opinion, the differences in microbial communities must be attributed mainly to environmental microbial contamination. As I previously stated in one of my comments, “The differences in the microbial groups found in raw milk should be attributed mainly to external or environmental contamination, and it should not be assumed that each sheep breed produces a characteristic milk with a specific microbiota.” Following the same reasoning, each individual of the same sheep breed could have a different “milk core microbiome”. The authors respond to this question as follows: “when secreted from a healthy udder the milk is almost sterile”, and they rely on two bibliographical references. Below I transcribe the last sentence as a conclusion from the first of these cited articles (P. Rainard, Vet Res 2017 48:25, DOI 10.1186/s13567-017-0429-2):

“Although the existence and importance of a teat apex microbiota deserves attention, it is the opinion of the author that the existence of an intramammary microbiota is a fiction that could cause confusion and interfere with practices that have proved useful for mastitis control.”

If it were assumed that microorganisms can cross the blood-mammary barrier and access the mammary alveoli, the application of hygienic measures to limit microbial contamination would be called into question. Based on the above, I think it is correct to speak about “raw milk microbiota” and relate it to an environment and livestock management conditions, but I do not consider the expression “milk core microbiome” to be justified in this case. Of course I agree with the conclusion that the authors point out in one of their answers as “the occurrence of specific taxa in samples collected from farms of the same geographic area supported the hypothesis that the environment shapes the milk microbiota”, but I do not see that hypothesis reflected in any paragraph of the manuscript. Finally, the authors do not make any mention of the sensory characteristics of the cheeses obtained in the two productions, nor do they even comment on whether their sensory quality is different.

Detailed comments

Title

- Change to e.g. “Microbiota diversity in raw milk from two farms managing two different breeds of sheep (Comisana and Lacaune)…”

Introduction

- Line 35. The reference with the number 7 refers to “extra-hard varieties”.

- Line 36. It is necessary to specify a range for the termization time. “At least 15 seconds” could be 15 hours. Change “for at least 15 seconds” to “for 15-… s”.

- Line 38. Change to “The cooking of the curd … at a temperature of 45-48 ºC”.

Materials and Methods

- Line 82. What is the meaning of “scotta-innesto”? Briefly explain or provide a reference.

Author Response

Responses to Reviewer 1’s comments

The Authors thank the Reviewer for the constructive and helpful comments. Here is a point-by-point response to the comments and concerns.

Comment: “Core microbiome”

Response: We appreciated the general comments on “the core microbiome,” and to avoid any misleading interpretation, we modified “core microbiome” with “shared taxa between the two datasets.” The concept of sheep core microbiome was introduced by Esteban-Blanco in 2019 (https://doi.org/10.1111/jbg.12446): “The core microbiota of the sheep milk includes five genera: Staphylococcus, Lactobacillus, Corynebacterium, Streptococcus and Escherichia/Shigella”. Comparing our data with those reported by these authors on the Assaf sheep breed, we demonstrated that the “Staphylococcus genus is not part of the core genera of the sheep milk microbiome” and can be considered an environmental contaminant.

Comment: “Milk sterility”

Response: As reviewed by several authors, the milk microbiota in mammals (humans and cows) contains both skin and environmental-derived microorganisms and bacterial species “emanating from the maternal gut to the mammary glands” (Singh et al., 2023, https://doi.org/10.1186/s12967-023-04656-9; Duale et al., 2021, https://doi.org/10.3389/fnut.2021.800927; Fernandez et al., 2013, https://doi.org/10.1016/j.phrs.2012.09.001). Based on the evidence that the “entero-mammary pathway” plays a crucial role in the structure of the milk microbiome in healthy mammals, studying the core genera microbiome can provide valuable information on the milk microbial community of different species.

Comment: “Sensory characteristics”

Response: The sensory analysis can provide valuable information on the characteristics of the final product, and we aim to include it in a future study. However, the cheese-making processes should be run independently on the same farm for a proper correlation between the sheep breed microbiome and the sensory cheese properties. In this work, we were interested in analyzing the changes in the pecorino-like cheese microbiome occurring in the artisanal process from two farms managing the milk of two different breeds of sheep.

Comment: Change to, e.g., “Microbiota diversity in raw milk from two different breeds of sheep (Comisana and Lacaune)…”

Response: The title was changed to “Microbial evolution in artisanal Pecorino-like cheeses produced from two farms managing two different breeds of sheep (Comisana and Lacaune).

Comment: Line 35. The reference with number 7 refers to “extra-hard varieties.”

Response: we have changed the reference; the new number 7 is “13.       Di Cagno, R.; Banks, J.; Sheehan, L.; Fox, P. F.; Brechany, E. Y.; Corsetti A.; Gobbetti M. Comparison of the microbiological, compositional, biochemical, volatile profile and sensory characteristics of three Italian PDO ewes’ milk cheeses. Int. Dairy J. 2003, 13, 961–972.”

Comment: Line 36. It is necessary to specify a range for the thermization time. “At least 15 seconds” could be 15 hours. Change “for at least 15 seconds” to “for 15-… s”.

Response: The time of thermization was specified as suggested (Line 36)

Comment: Line 38. Change to “The cooking of the curd … at a temperature of 45-48 ºC”.

Response: The text has been revised as suggested by the reviewer (Line 37)

Comment: Line 82. What is the meaning of “scotta-innesto”? Briefly explain or provide a reference.

Response: The meaning of the term “scotta-innesto” was explained as suggested

Line 112-113, “residual whey starter cultures from ricotta cheese manufacturing (scotta-innesto)”

Round 3

Reviewer 1 Report

Comments and Suggestions for Authors

Overview and comments

The authors have responded appropriately and kindly to my questions. Some of the concepts handled in the study have been clarified and suitable modifications have been made. In particular, I think that the new title of the article is more appropriate and avoids a more than questionable assumption that the “breed effect” determines the composition of the microbiota found in the production of the two farms, especially the microorganisms that could show technological interest. I hope that the authors will carefully and rigorously review the final version of the article and correct any errors that may have been omitted. I also trust the criteria and responsibility of the authors when recording and transferring all the results.

Finally, I think there is a minor correction that should be made in the Materials and Methods section:

-Line 84 (?). Specify that Ricotta cheese is a whey cheese (I assume it is made with the whey obtained in the manufacture of Pecorino-like cheese), since many readers may not know this particular fact. It is also advisable to write “scotta-innesto” in italics. Change to e.g.: “residual whey native starter cultures from Ricotta (whey) cheese manufacturing (scotta-innesto)”.

Author Response

Responses to Reviewer 1’s comments

Comment: Line 84 (?). Specify that Ricotta cheese is a whey cheese (I assume it is made with the whey obtained in the manufacture of Pecorino-like cheese), since many readers may not know this particular fact. It is also advisable to write “scotta-innesto” in italics. Change to e.g.: “residual whey native starter cultures from Ricotta (whey) cheese manufacturing (scotta-innesto)”.

Response: The text has been revised as suggested by the reviewer (Line 119-120)